# Molecular Diversity and Phylogeny Reconstruction of Genus *Colobanthus* (Caryophyllaceae) Based on Mitochondrial Gene Sequences

**DOI:** 10.3390/genes13061060

**Published:** 2022-06-14

**Authors:** Piotr Androsiuk, Łukasz Paukszto, Jan Paweł Jastrzębski, Sylwia Eryka Milarska, Adam Okorski, Agnieszka Pszczółkowska

**Affiliations:** 1Department of Plant Physiology, Genetics and Biotechnology, Faculty of Biology and Biotechnology, University of Warmia and Mazury in Olsztyn, ul. M. Oczapowskiego 1A, 10-719 Olsztyn, Poland; bioinformatyka@gmail.com (J.P.J.); sylwia.milarska@uwm.edu.pl (S.E.M.); 2Department of Botany and Nature Protection, Faculty of Biology and Biotechnology, University of Warmia and Mazury in Olsztyn, ul. Prawocheńskiego 17, 10-720 Olsztyn, Poland; pauk24@gmail.com; 3Department of Entomology, Phytopathology and Molecular Diagnostics, Faculty of Agriculture and Forestry, University of Warmia and Mazury in Olsztyn, ul. Prawocheńskiego 17, 10-720 Olsztyn, Poland; adam.okorski@uwm.edu.pl (A.O.); agnieszka.pszczolkowska@uwm.edu.pl (A.P.)

**Keywords:** mitogenome, nucleotide substitutions, RNA editing, phylogeny, NGS

## Abstract

Mitochondrial genomes have become an interesting object of evolutionary and systematic study both for animals and plants, including angiosperms. Although the framework of the angiosperm phylogeny was built on the information derived from chloroplast and nuclear genes, mitochondrial sequences also revealed their usefulness in solving the phylogenetic issues at different levels of plant systematics. Here, we report for the first time the complete sequences of 26 protein-coding genes of eight *Colobanthus* species (Caryophyllaceae). Of these, 23 of them represented core mitochondrial genes, which are directly associated with the primary function of that organelle, and the remaining three genes represented a facultative set of mitochondrial genes. Comparative analysis of the identified genes revealed a generally high degree of sequence conservation. The Ka/Ks ratio was <1 for most of the genes, which indicated purifying selection. Only for *rps12* was Ka/Ks > 1 in all studied species, suggesting positive selection. We identified 146–165 potential RNA editing sites in genes of the studied species, which is lower than in most angiosperms. The reconstructed phylogeny based on mitochondrial genes was consistent with the taxonomic position of the studied species, showing the separate character of the family Caryophyllaceae and close relationships between all studied *Colobanthus* species, with *C. lycopodioides* sharing less similarity.

## 1. Introduction

Caryophyllaceae is a family of flowering plants which includes 81 genera and about 2625 species [1]. Most of them are herbaceous plants found in temperate climates, with many ornamental plants or widespread weeds among them. However, there are also some taxa with very unique distribution patterns or unusual characteristics, such as the genus *Colobanthus*. Currently, 26 species can be distinguished within the genus *Colobanthus* [2], which can be found mostly in the Southern Hemisphere including the Pacific Region, Australasia, southern South America, sub-Antarctic islands, and the maritime Antarctic, but the major diversity is located in New Zealand [3]. Genus *Colobanthus* contains perennial herbs with dense and narrow leaves, cushion growth habit, and small pentamerous, greenish or white flowers without petals but with prominent sepals [4,5]. Some of the *Colobanthus* representatives, such as *C. affinis*, *C. apetalus*, or *C. quitensis* are widespread species, with the latter considered as the most wide-ranging species within the genus, occurring from the Antarctic Peninsula [6] north along the southern Andes mountains up to scattered locations northward to Mexico [7,8]. On the other hand, among *Colobanthus* species, there are also examples of very rare and endangered taxa which can be found on the list of Australian rare or threatened plant species [9]: for example, *C. nivicola*, which is endemic to Mt Kosciusko area in New South Wales, Australia [10], or *C. curtisiae*, *C. strictus*, and *C*. *squarrosus*, which were recorded from only three populations located in Tasmania [3,11,12]. *Colobanthus* species are often characterized by high phenotypic plasticity, which arises in response to different environmental conditions [13]. Such ecotypic differentiation is a common phenomenon among plant populations, especially when they are exposed to various stress factors [14]. Additionally, in the case of sympatric species, difficulties in species identification are possible due to high morphological similarity, as was reported for example for *C. nivicola* and *C. pulvinatus* [10].

Mitochondria are uniparentally inherited organelles whose main role is the production of cellular ATP during the process of oxidative phosphorylation. Although mitochondria contain their own genetic material (mitochondrial genome), they are called semi-autonomous organelles, as many of their functions are regulated by genes encoded by the nuclear genome [15]. The plant mitochondrial genome differs significantly in size, composition, and structural organization from the mitogenome of other eukaryotic organisms [16]. The plant mitochondrial genome usually is a double-stranded, circular molecule, although several independent chromosomes [17] or linear and multi-branched structures have been also reported [18]. Its size varies significantly from 66 Kbp (*Viscum scurruloideum*; [19]) up to 11.7 Mbp (*Larix sibirica*; [20]), and the observed differences are largely caused by variation in the length of non-coding intergenic regions and presence/absence of introns [21,22,23]. The structure of plant mitogenome, during its evolution, experienced many rearrangements associated with gene duplications or gain/loss phenomena [24,25]. Furthermore, the incorporation of a number of foreign sequences of nuclear and chloroplast origin [26,27], as well as the presence of sequences from the mitochondrial genomes of other plant species acquired via horizontal transfer [28,29] shows that plant mitogenomes often undergo genome recombination. Additionally, the presence of a large number of various repeated sequences promotes further changes in genomic configuration [30,31]. As a consequence, the sequencing and assembly of the plant mitochondrial genome is more demanding than in the case of animal mitogenomes or even chloroplast genomes. Nevertheless, there has been a continuous rise in the scientific interest in plant mitochondrial genomes, as they may have a great value for plant breeding as a source of molecular markers [32] or due to the fact that mitogenomes contain genes responsible for cytoplasmatic male sterility, i.e., the genes which are involved in the production of functional pollen or proper functioning of male reproductive organs [33,34]. Additionally, slow sequence evolution, generally the slowest among the three plant genomes when the sequence level is considered, and variation associated with introns make the plant mitogenome an interesting source of phylogenetic information [35]. Unfortunately, the possible potential of plant mitochondrial genomes is not fully explored due to the limitations associated with sequencing technology. Although next-generation sequencing (NGS) technology is currently used worldwide to determine the order of nucleotides in targeted regions of DNA or entire genomes, it is not free of constraints, among which the single read length is the main limitation. Nevertheless, the development of long-read sequencing technologies offered by Pacific Biosciences (PacBio) and Oxford Nanopore Technologies (ONT) appeared as a tool that will help the researchers to overcome that constraint.

Currently, there are 11,769 animal mitochondrial genomes deposited in the Organelle Genome Resources database at the NCBI server (https://www.ncbi.nlm.nih.gov/genome). However, there are only 433 plant mitogenomes, out of which 319 represent land plants (data valid for February 2022). Within the family Caryophyllaceae, there are three complete and annotated mitochondrial genomes for *Silene latifolia*, *Silene vulgaris*, and *Agrostemma githago*, but when the searching is expanded to the whole order Caryophyllales, one can find an additional 14 mitogenomes for *Beta macrocarpa*, *Beta vulgaris* subsp. *maritima*, *Beta vulgaris* subsp. *vulgaris*, *Bougainvillea spectabilis*, *Chenopodium quinoa*, *Mirabilis himalaica*, *Mirabilis jalapa*, *Nepenthes ventricosa × Nepenthes alata*, *Spinacia oleracea*, *Suaeda glauca*, *Sesuvium portulacastrum*, *Fallopia aubertii*, *Fallopia multiflora*, and *Tetragonia tetragonoides*. All of the above-mentioned mitochondrial genomes, except for the last four, are included in the RefSeq database which is a comprehensive, integrated, non-redundant, well-annotated set of reference sequences including genomic, transcript, and protein.

Genomic data for the genus *Colobanthus* are very limited. There are complete chloroplast genome sequences for eight representatives of that genus [36,37,38] and two transcriptome shotgun assemblies for *C. quitensis* [39,40,41]. Up to now, no mitogenome sequences of *Colobanthus* has been published. In this study, we report complete sequences of 26 mitochondrial protein-coding genes shared by *Colobanthus affinis*, *C. apetalus*, *C. curtisiae*, *C. lycopodioides*, *C. muscoides*, *C. nivicola*, *C. pulvinatus* and *C. quitensis*. Using the sequence data for eight *Colobanthus* representatives, the interspecific comparative analyses were performed to study the evolution of functional mitochondrial genes and their scale. Furthermore, the phylogenetic relationships between the studied *Colobanthus* species and their position within order Caryophyllales were also estimated based on mitogenomic data.

## 2. Materials and Methods

### 2.1. Plant Material, DNA Extraction and Mitochondrial Genome Sequencing

The research material included fresh leaves of eight *Colobanthus* species sampled from plants grown from seeds in a greenhouse of the Department of Plant Physiology, Genetics and Biotechnology at the University of Warmia and Mazury in Olsztyn, Poland. The seeds of *C. nivicola* and *C. pulvinatus* were acquired from the Australian National Botanic Gardens, Canberra. The seeds of *C. affinis* were obtained from the Royal Botanic Gardens, Victoria, Australia. The seeds of *C. curtisiae, C. muscoides* originated from the Royal Botanic Gardens, Kew, United Kingdom. The seeds of *C. lycopodioides* were collected in the region of Mendoza, Andes, Argentina, at an altitude of 4024 m a.s.l., (33 10′ S; 69 50′ W). *C. apetalus* seeds were collected on the southeastern shore of Lago Roca, near Lapataia Bay, Tierra del Fuego, Argentina. The *C. quitensis* originated from King George Island (South Shetland Islands) were collected in the vicinity of Henryk Arctowski Polish Antarctic Station. Professor Irena Giełwanowska performed morphological and anatomical analyses of both vegetative and generative organs harboring characteristic traits for the identification of *Colobanthus* species [2,42,43]. Voucher specimens of studied species have been deposited in the Vascular Plants Herbarium of the Department of Botany and Nature Protection at the University of Warmia and Mazury in Olsztyn, Poland (OLS), under the following numbers: *C. affinis* (No. OLS 33825), *C. apetalus* (OLS33831), *C. lycopodioides* (No. OLS 33826), *C. nivicola* (No. OLS 33827), *C. pulvinatus* (No. OLS33828), and *C. quitensis* (No. OLS33833). Only in case of *C. curtisiae* and *C. muscoides*, due to their low growth rate and lack of flowering, no voucher specimen has been deposited. The verification of these species was performed by the seeds contractor i.e., Royal Botanic Gardens, Kew, United Kingdom (the written confirmation was supplied with the seeds).

Total genomic DNA was extracted from fresh tissue of a single individual with the application of the Maxwell16LEV Plant DNA Kit (Promega, Madison, WI, USA). The quality of DNA was verified on 1% (*w*/*v*) agarose gel stained with 0.5 μg/mL ethidium bromide. The concentration and purity of DNA samples were assessed spectrophotometrically. Genome libraries were prepared using the Nextera XT kit (Illumina Inc., San Diego, CA, USA) and were sequenced on the Illumina MiSeq platform (Illumina Inc., San Diego, CA, USA) with a 150 bp paired-end read.

### 2.2. Gene Annotation and Comparative Analysis

The quality of the obtained raw reads was evaluated with the FastQC tool. All raw reads were trimmed (the reads with PHRED score <20 were removed), and next, clean reads from *C. affinis* were mapped to the reference mitochondrial genome of *Silene vulgaris* (JF750427) using default medium-low sensitivity parameters of Geneious Prime v.22.1.1 software [44]. The mapped CDS regions were extracted and remapped by the *C. affinis* reads with high sensitivity: minimum overlap >50 and minimum overlap identity >96%. The remapping process was used to elongate extracted contigs and was looped until complete coding sequences were identified. The assembly coding genes were checked by acceptor–donor sites identification, length similarity, and homology parameters using standalone Megablast software [45]. Only intact, complete copies of genes were included in subsequent analyses; therefore, the lack of a particular gene in this set of coding sequences should not be treated as a proof of its absence in studied mitogenomes. The obtained 26 *C. affinis* coding sequences were used as a reference to the mapping procedure for the remaining seven *Colobanthus* species. Each library for the *Colobanthus* species was remapped to *C. affinis* CDS references with medium–low sensitivity.

A comparative analysis of reported mitochondrial protein-coding genes of eight *Colobanthus* species (*C. affinis*, *C. apetalus*, *C. curtisiae*, *C. lycopodioides*, *C. muscoides*, *C nivicola*, *C. pulvinatus*, and *C. quitensis*) included codon usage analysis, estimation of synonymous and non-synonymous substitutions and the detection of potential RNA editing sites.

The evolutionary rate of the mitochondrial genes identified in studied *Colobanthus* species was also studied. A total of 26 genes were analyzed to estimate the ratio of non-synonymous (Ka) to synonymous (Ks) substitutions. *Colobanthus quitensis* was the reference species. These genes were extracted and aligned separately using MAFFT v7.310 [46]. The values of Ka and Ks in the shared genes were calculated in DnaSP v.6.10.04 [47]. Genes with non-applicable (NA) Ka/Ks ratios were changed to zero.

The potential RNA editing sites in all 26 protein-coding genes in the mitochondrial genome of eight *Colobanthus* species were predicted using the Predictive RNA Editor for Plants (PREP) suite [48] and Deepred-Mt tool [49]. During the analyses, the cut-off value was set at 0.2, which is a default setting in the PREP suite for mitochondrial sequences. The same threshold value was applied for Deepred-Mt. Only potential RNA editing sites predicted by both tools were reported.

### 2.3. Phylogenetic Analysis

In order to check for the suitability of our mitochondrial genome data for phylogeny reconstruction, we have applied all of the 26 sequences of mitochondrial protein-coding genes shared by eight *Colobanthus* species reported in this paper as well as genomic data for other representatives of order Caryophyllales for which the complete mitochondrial genomes were available in GenBank (NCBI). Additionally, the *Arabidopsis thaliana* was used here as an outgroup. All appropriate sequences were downloaded from the NCBI database. Initially, protein-coding genes which were identified for analyzed *Colobanthus* species were searched for and extracted from GenBank records of mitochondrial genomes of selected Caryophyllales representatives. For this purpose, a custom R script was used. As a result, we obtained sequences of 18 protein-coding genes (*atp1*, *atp6*, *atp9*, *ccmB*, *cob*, *cox1*, *cox2*, *cox3*, *matR*, *nad1–7*, *nad4L*, and *nad9*), which were shared by eight *Colobanthus* species, 11 representatives of order Caryophyllales, and *A. thaliana* (Table 1). In the next step, 18 concatenated protein-coding gene sequences for the above-mentioned set of 20 species were aligned in MAFFT v7.310 and used for phylogeny reconstruction by Bayesian Inference (BI) and Maximum-Likelihood (ML) method. The BI analysis was performed in MrBayes v.3.2.6 [50,51], and the ML analysis was conducted in PhyML v.3.0 [52], using the parameter settings described previously [37]. However, before BI and ML analysis, the best fitting substitution model was searched in MEGA 7 [53], and the model GTR + G was selected for both variants of analysis. Simultaneously, the second variant of phylogenetic analysis was conducted, in which an investigation of phylogenetic relationships only among the studied representatives of the genus *Colobanthus* was performed based on the whole set of shared 26 protein-coding genes. In addition, in this case, the gene sequences were aligned in MAFFT v7.310, and BI and ML methods were used for phylogeny reconstruction. The second variant of phylogenetic studies was performed to test how, and to what extent, the reduced number of genes (18) applied in the first variant of analysis might modify the phylogenetic relationship revealed between studied *Colobanthus* species based on a whole set of 26 genes. The results of that comparison will enable us to evaluate the suitability of the set of 18 mitochondrial genes for the reconstruction of phylogenetic relationships between selected representatives of order Caryophyllales.

## 3. Results

### 3.1. Mitochondrial Genes Assembly, Annotation and Characteristics

After the sequencing of our eight *Colobanthus* species, the following numbers of raw reads were obtained: 4,940,388 (*C. affinis*), 22,967,202 (*C. apetalus*), 24,320,294 (*C. curtisiae*), 5,780,642 (*C. lycopodioides*), 21,713,178 (*C. muscoides*), 17,366,398 (*C. nivicola*), 5,473,888 (*C. pulvinatus*), and 29,777,488 (*C. quitensis*). Trimmed reads of the low abundance DNA-seq library (*C. affinis*) were mapped to *Silene vulgaris* mitochondrial genome, and 26 protein-coding genes were covered on full sequence length. These *Colobanthus* reference mitochondrial genes were simultaneously mapped by libraries of the other seven *Colobanthus* species. All 26 sequences were identified, and the remapping rate for each species group was respectively: 4049 uniquely mapped reads (*C. affinis*), 9260 (*C. apetalus*), 8846 (*C. curtisiae*), 5361 (*C. lycopodioides*), 14,181 (*C. muscoides*), 13,130 (*C. nivicola*), 4120 (*C. pulvinatus*), and 18,983 (*C. quitensis*). The sequencing data obtained during the project realization were deposited in GenBank (NCBI) under project accession PRJEB52939. Additionally, sequences of all identified protein-coding genes for each *Colobanthus* species were attached to the manuscript as Appendix A.

Our sequencing data allowed us to obtain the complete coding sequences of 26 mitochondrial genes in all the studied *Colobanthus* species, and only these sequences were subjected to further analyses. Among these sequences, there were genes for four subunits of ATP synthase (atp1,atp4, atp6, and atp9), four genes associated with cytochrome c biogenesis (ccmB, ccmC, ccmFc, and ccmFn), a gene for ubiquinol cytochrome c reductase (cob), three genes for subunits of cytochrome c oxidase (cox1–3), a maturase gene (matR), a sequence for transport membrane protein (mttB), sequences for nine subunits of NADH dehydrogenase (nad1–6, nad7, nad9, and nad4L) and sequences for three components of the small subunit of the ribosome (rps11–13) (Appendix A). All genes appeared to be highly conservative when their size was considered. However, in the case of *C. lycopodioides*, seven genes (*ccmFn*, *cox2*, *matR*, *mttB*, *nad6*, *nad7*, and *rps12*) revealed differences in their length, which arose as a consequence of deletion and/or insertion events that caused contraction (*matR*, *mttB*, *nad6*, *nad7*, *rps12*) and less often elongation (*ccmFn* and *cox2*) of the total gene length (Appendix A). Alignment of the final, complete sequences of each of the 26 reported here protein-coding genes for all studied *Colobanthus* species revealed that the highest number of indels was observed for *rps12*, *matR* and *ccmFn* genes (File S9). Three insertions and two deletions were observed in the alignment of *rps12* gene sequences for all eight *Colobanthus* species. This indel polymorphism shortened the total length of the *rps12* sequence in the mitochondrial genome of *C. lycopodioides* by nine nucleotides in comparison to other *Colobanthus* species. In the case of the *matR* gene, three deletions truncated its sequence by 15 nucleotides in total, whereas two insertions and one deletion increased the total sequence length of *ccmFn* by three nucleotides. For the remaining genes, one or two indels were identified in each of them. In the case of *cox2*, the elongation of that gene (insertion of six nucleotides) was observed not only in the mt genome of *C. lycopodioides* but also in *C. quitensis* and *C. curtisiae*.

The majority of identified genes started with the typical AUG codon and terminated with UAA, UGA, or UAG (Appendix A). The UAG termination codon was observed for only three genes, i.e., *atp4*, *matR*, and *rps12*, but with one exception for the latter for *C. lycopodioides* for which a UAA codon was observed. In the case of the *mttB* gene, an AUA initiation codon was observed, whereas for *nad1* and *nad4L*, an ACG codon was found at the beginning of the coding sequence. The total number of codons or reported protein-coding genes in the mitochondrial genomes of studied *Colobanthus* species ranged from 8154 for *C. lycopodioides* to 8168 for *C. curtisiae* and *C. quitensis*. The most and least abundant codons (excluding these associated with the initiation and termination of translation) were TTT (on average 4.81%) and TGC (on average 0.37%), respectively (Appendix A). Moreover, leucine appeared as the dominant amino acid (on average 12.1%), whereas cysteine was less frequently encountered (on average 1.33%).

### 3.2. Synonymous (Ks) and Non-Synonymous (Ka) Substitution Rate Analysis

The analysis of the substitution rate across the reported mitochondrial genes revealed that it varied widely in each functional group and across the studied species. The value of Ka and Ks were estimated, and they were in the following ranges: 0–0.1333 and 0–0.4757 for Ka and Ks, respectively (Appendix A, Figure 1A,B). The highest average values of Ks (0.1260) and Ka (0.0332) were observed for the same group of coding sequences, i.e., genes of subunits of cytochrome c oxidase. The average value of Ks was the lowest (0.0053) in genes related to cytochrome c biogenesis. The same group of genes was also characterized by the lowest average Ka value (0.0022). Similarly low values of average Ka were observed also for genes associated with ATP synthase (0.0024) and NADH dehydrogenase (0.0032). When each gene was analyzed individually, the highest average Ks value (0.3520) was found for *cox2*, which was followed by *atp9* with an average Ks = 0.1281 (Appendix A). The lowest average Ks was found for *ccmC* (0.0008). When the non-synonymous substitution rates (Ka) were considered, the highest average value of that parameter was found for *cox2* and *rps12* (0.0975 and 0.0459, respectively), whereas the lowest were found for *cox1*, *rps11*, and *atp6* (0.0004, 0.0005 and 0.0007, respectively).

There were five genes (*ccmB*, *ccmFn*, *cob*, *nad4L*, and *rps13*) for which no differences were observed in their sequences (Ks = 0, Ka = 0) in seven out of eight of the analyzed *Colobanthus* species (*C. lycopodioides* was this exception). There were also six genes (*cox1*, *nad1*, *nad3*, *nad6*, *nad7*, and *rps11*) for which only synonymous substitutions were observed (Ka = 0), and seven other genes (*atp4*, *atp6*, *ccmC*, *matR*, *mttB*, *nad4* and *nad9*) for which only non-synonymous substitutions were scored (Ks = 0). The only exception from these rules was *C. lycopodioides*, which differs from all other species in Ka and Ks values estimated for all genes, except *nad1* for which Ka = 0 was observed for all studied species including *C. lycopodioides*. The Ks values for *C. lycopodioides* ranged from 0.0056 to 0.34, whereas Ka values were within the range 0–0.1223. The Ka/Ks ratio was also estimated. For most genes, the Ka/Ks ratio was <1, which indicated purifying selection. In the case of the *atp1* gene from the mitogenome of *C. nivicola* and *C. pulvinatus*, Ka/Ks = 1, which indicated neutral selection. Only for *rps12* was the Ka/Ks ratio > 1, suggesting positive (i.e., driving change) selection in all studied species (Appendix A, Figure 1C).

### 3.3. RNA Editing Sites

The application of PREP suite and Deepred-Mt programs allowed us to predict from 146 to 165 RNA editing sites in analyzed protein-coding genes in mitogenomes of *Colobanthus* species. Out of all 26 genes for which complete coding sequences were obtained, we did not find potential RNA editing sites in *atp9* and *cox1* in all *Colobanthus* species and in *cox2* in the case of *C. curtisiae* and *C. quitensis*. All of the identified potential RNA editing sites were C-to-U conversion in the first or second position of the codon. The second position of the codon was altered more often, as the C-to-U edition was observed there with the frequency of 61.2–61.6%. The lowest number of RNA editing sites (146) was revealed for *C. quitensis* and *C. curtisiae* (Appendix A), whereas for *C. apetalus*, *C. affinis*, *C. muscoides*, *C. nivicola*, and *C. pulvinatus* 148 of such elements were identified (Appendix A). The discrepancy between these two groups of species was associated with the lack of potential RNA editing sites in *cox2* sequence in the case of *C. curtisiae* and *C. quitensis*. In the case of *C. apetalus*, *C. affinis*, *C. muscoides*, *C. nivicola*, and *C. pulvinatus*, two potential RNA editing sites were reported for *cox2*: modification, which altered the codon for serine to leucine (TCA (S) => TTA (L)) and arginine to tryptophan (CGG (R) => TGG (W)). In the case of *C. lycopodioides*, we revealed 165 potential RNA editing sites (Appendix A). This inconsistency in comparison to other *Colobanthus* species was associated with the different numbers and positions of potential RNA editing sites in sequences of thirteen genes: *atp6*, *ccmFn*, *cob*, *cox2*, *matR*, *mttB*, *nad2*, *nad3*, *nad4L*, *nad5*, *nad7*, *nad9,* and *rps12*. Generally, when RNA editing sites for all *Colobanthus* species were considered, the serine (S) to leucine (L) change was the most frequent; it accounted for 29.1–31.5% of the identified mutations, and it was followed by serine (S) to phenylalanine (F) and proline (P) to leucine (L) changes (12.8–13.9%, in both cases). On the other hand, threonine (T) to methionine (M) (0.7–1.2%)) and proline (P) to phenylalanine (F) (1.4–1.8%) were least frequently observed. The highest number of potential RNA editing sites in one gene was identified within sequences for *ccmB* (19) and *nad5* (17) in the case of all *Colobanthus* species. Additionally, within the sequence of *ccmFn*, 17 potential editing sites were recognized for *C. lycopodioides*. Only one potential RNA editing site was found within *cox3* and *rps11* in all analyzed species as well as in *cob* and *nad3* in all *Colobanthus* species except *C. lycopodioides*. Finally, one potential RNA editing site was identified in *cox2* but only in *C. lycopodioides*.

### 3.4. Phylogenetic Analysis

The application of BI and ML methods resulted in the generation of phylogenetic trees which had a consistent topology in both variants of the analysis. In the first variant, the BI tree posterior probability reached the value of 1.0 in almost all nodes (15 out of 17; 88.2%) (Figure 2). Analysis of the phylogenetic tree revealed that all *Colobanthus* species were gathered in one clade; this was closely related to the second clade, which consisted of three species (*A. githago* and a pair of *Silene* species, i.e., *S. latifolia* and *S. vulgaris*). The third clade included *B. macrocarpa*, *B. vulgaris* subsp. *maritima* and *C. quinoa*. The fourth clade consisted of *S. portulacastrum*, *T. tetragonoides* and *M. himalaica*. *N. ventricosa* × *N. alata* and *F. aubertii* form together the fifth clade. Finally, the most divergent branch was formed by the *A. thaliana* (outgroup). The second variant of phylogenetic analyses included only eight *Colobanthus* species. In addition, in this case, very high BI tree posterior probability values were observed: the value of 1.0 in four out of five nodes and 0.996 in the remaining one (Figure 3)**.** Analysis of the tree revealed that *C. pulvinatus* and *C. nivicola* share the highest similarity. Subsequently, *C. apetalus*, *C. muscoides* and *C. affinis* joined the mentioned above pair of species and were followed by *C. curtisiae* and *C. quitensis* (the order of the species name listed above reflect the descending similarity between them). *C. lycopodioides* occupied the most divergent branch out of all *Colobanthus* species.

## 4. Discussion

According to the available literature, the number of protein-coding genes in angiosperm mitogenomes varies considerably and may range from 32 to 67 [54]. Richardson et al. [55] based on sequence data for the mitochondrial genome of *Liriodendron tulipifera* anticipated that ancestral flowering plant mitochondria contained 41 protein-coding genes, among which 24 were considered as highly conservative elements of this organellar genome, whereas another 17 (including ribosomal *rpl* and *rps* and succinate dehydrogenase (*sdh)* genes) were regarded as facultative for angiosperm mitogenomes [56].

There are three complete, fully annotated mitochondrial genomes (for *Silene latifolia*, *S. vulgaris*, and *Agrostemma githago*) within the Caryophyllaceae family. The gene composition of all of these genomes is very conservative with an obligatory set of 24 core protein-coding genes (which include *atp1*, *atp4*, *atp6*, *atp8*, *atp9*, *ccmB*, *ccmC*, *ccmFc*, *ccmFn*, *cob*, *cox1–3*, *matR*, *mttB*, *nad1–7*, *nad9*, and *nad4L*) and various set of *rpl*, *rps*, and *sdh* genes. Additionally, a sequence similar to the DNA polymerase (*dpo*) was identified for *S. latifolia* and *S. vulgaris*. This observation is convergent with the results of previous studies which showed that the loss of genes was not a rare phenomenon in angiosperm mitogenomes and generally affected different genes representing *rpl*, *rps*, and *sdh* genes [57,58]. Moreover, RNA and DNA polymerase genes were also reported in the mitochondrial genomes of other angiosperms [59].

In the current study, an analysis of NGS data for mitochondrial genomes of eight representatives of the genus *Colobanthus* allowed us to obtain for the first time complete sequences for 26 protein-coding genes. So far, no mitochondrial sequences for any *Colobanthus* species were available in GenBank (NCBI). Twenty-three of the reported gene sequences represented the above-mentioned set of core mitochondrial protein-coding genes, which are nearly universally conserved across angiosperm mitochondrial genomes, whereas the subsequent three (*rps11*, *rps12*, and *rps13*) represent sequences for proteins of the small subunit of the ribosome. Comparative analysis of the identified mitochondrial genes revealed that they generally maintain a high degree of sequence conservation. It was manifested by the presence of genes (*ccmB, ccmFn, cob, nad4L,* and *rps13*) for which no differences were observed in the studied species as well as genes for which only synonymous substitutions were identified (*cox1*, *nad1*, *nad3*, *nad6*, *and7*, and *rps11*). Moreover, even when non-synonymous substitutions were observed, the Ka values showed rather low values, which reach the maximum value of 0.1333 for *cox2* for most of the *Colobanthus* species and 0.1223 for *rps12* for *C. lycopodioides,* but for the remaining genes, it did not exceed 0.0605. Only genes of *C. lycopodioides* did not follow the above-mentioned rules, which resulted in the most distinct character of the species, which was reflected in the values of synonymous and non-synonymous substitution rates.

According to expectations, synonymous nucleotide substitutions in studied mitochondrial genes of eight *Colobanthus* species were more frequent than non-synonymous substitutions [60]. Moreover, the observed substitution rates appeared to be minor and consistent with low rates that generally characterize the plant mitochondrial genomes [61,62]. Nevertheless, we have identified two genes for which an increased synonymous substitution rate was observed: the *atp9* and *cox2*. In our study, the Ks values for *atp9* sequences ranged from 0.0728 (for *C. apetalus* and *C. muscoides*) up to 0.2201 (for *C. lycopodioides*) giving on average 0.1281, which is the second-largest value of this parameter in our data set, although *atp9* appeared as the shortest (225 nt) gene among all reported here for the mitochondrial genome of *Colobanthus* species. The highest Ks values were observed for *cox2*, and they reach the value of 0.4757 for most of the species. Only for *C. curtisiae* and *C. lycopodioides*, the alignment of *atp9* gene with the corresponding sequence in the mitogenome of *C. quitensis* resulted in lower Ks values of 0 and 0.0852, respectively. Multiple alignment of *cox2* sequences for all studied *Colobanthus* species showed a total of 147 substitution sites and one insertion of 6 nt (CGACAC coding arginine and histidine) in *C. quitensis*, *C. curtisiae* and *C. lycopodioides*, which resulted in an extension of the sequence of this gene to 780 nt. Although *atp9* and *cox2* were characterized by the accelerated substitution rate, they still revealed evidence of purifying selection (Ka/Ks ratio <1) and absence of premature internal stop codons, which suggests that they are functionally expressed in the mitochondria.

Accelerated substitution rates for *atp9* have been previously reported also for *Silene* [62], *Geranium* [63], or *Boehmeria nivea* [64]. In the case of *cox2*, there is a very limited number of research that points to the accelerated substitution rate of that gene. Such phenomena were described, e.g., for mitochondrial genomes of two *Geranium* species, *G. brycei* and *G. incanum* [62]. In the case of the above-mentioned *atp9* gene in *Silene* and *Geranium*, the authors pointed to the acceleration of mutation rate as the most probable explanation of elevated substitution rate [62,63]. In the case of *B. nivea*, the authors concluded that a higher substitution rate of the *atp9* could be a reason for the cytoplasmic male sterility and strong environmental adaptations observed for this species. For *cox2*, the observed elevation of substitution rate in *Geranium* species was explained by the effect of the conversion between the *cox2* copies acquired by these mitogenomes probably via the horizontal gene transfer [63,65]. Synonymous substitutions, due to the fact that they do not affect the corresponding amino acid sequence, were considered for a long time to be neutral, i.e., with no fitness effect [66,67]. However, there are also studies that revealed that they could be subjected to various selection pressure including the efficiency of translation, mRNA stability, or the conservation of regulatory motifs [68]. Synonymous substitutions, as not completely neutral evolutionary forces, appeared then as an interesting object of evolutionary studies.

In order to infer the direction and magnitude of natural selection acting on reported protein-coding genes, the Ka/Ks ratio was also estimated. Generally, the Ka/Ks ratio was <1 for most of the genes, which indicated negative selection. However, we identified also two alternative strategies of gene evolution: in case of the *atp1* gene from the mitogenome of *C.nivicola* and *C. pulvinatus* Ka/Ks = 1, which indicated neutral selection, whereas for *rps12* in all studied species, the Ka/Ks ratio was >1, suggesting positive selection. The observed pattern of Ka/Ks values is concordant with the expectations concerning the mitochondrial genomes for which negative selection was reported to be the predominant force of evolution [69]. This functional constraint of mitochondrial genes is associated with the important role of the organelle which is sustained by purifying selection, which is responsible for maintaining the long-term stability of biological structures by the elimination of deleterious variations. As a consequence, protein-coding genes in the mitochondrial genome are conserved across land plants [70]. Positive selection is the process that fixes beneficial variations in the population and promotes the emergence of new phenotypes as an adaptation to certain environmental conditions. Analysis of the available literature showed that it is not often in mitochondrial genomes, but traces of that model of evolution were previously reported, e.g., in *ccmB, ccmFc*, and *mttB* genes in Lamiales [71], *ccmB* for *Suaeda glauca* [70] or *rps19* for *Cucumis* [72].

There is also another source of sequence variation in organelle genomes of land plants. This variation arises as a consequence of the RNA editing process, which involves pyrimidine exchange, mostly C-to-U, i.e., the conversion of cytidine to uridine by deamination [73]. Almost all edits are non-synonymous and cause an amino acid translation different from that encoded by the original genome sequence [74]. RNA editing is believed to be essential for organelle gene functions in plants [75] and is more frequent in mitochondrial genomes than in chloroplasts [76]. The clear disproportion in the number of RNA editing sites between chloroplast and mitochondrial genome was observed also for *Colobanthus* species. In our previous study, which reported the complete chloroplast genomes of six *Colobanthus* species [38], 49 potential RNA editing sites were identified within 18 out of the 34 analyzed chloroplast protein-coding genes. In the present study, potential RNA editing sites were identified in 23 (*C. curtisiae* and *C. quitensis*) or 24 (the other *Colobanthus* species) out of the 26 analyzed mitochondrial protein-coding genes. The number of RNA editing sites ranged from 146 (for *C. quitensis* and *C. curtisiae*), through 148 (for *C. apetalus, C. affinis, C. muscoides, C. nivicola, C. pulvinatus*) up to 165 (for *C. lycopodioides*), which is lower in comparison to the other representatives of Caryophyllaceae, i.e., *Silene noctiflora* and, *Silene latifolia* (189 and 287, respectively; [75]) or other angiosperms such as *Arabidopsis* (456; [73]). This decline in the number of RNA editing sites observed for *Colobanthus* species can be the result of a reduction of the number of mitochondrial genes in comparison to other angiosperm species such as *Arabidopsis,* or a lower average density of editing sites within each gene, when a comparison is made between more closely related species from Caryophyllaceae family. Despite the clear discrepancy between the number of potential RNA editing sites identified in protein-coding genes of chloroplast and mitochondrial genomes of *Colobanthus* species, they revealed very similar characteristics; i.e., all of them represented C-to-U conversion, were located predominantly at the second position of the codon, and most of them represented serine (S) to leucine (L) change. The generation of genetic variation and involvement in gene regulation are not the only effects of RNA editing sites. They are also involved in the preservation of different functional protein isoforms [77], nuclear control of selfish organelle genes [78], and mutational buffering [79]. As a consequence, the identification of RNA editing sites became an obligatory element not only of studies reporting the results of organelle genome sequencing but also in projects which ask evolutionary questions concerning the origin and function of various elements of gene and genome architecture [80].

Since the 1990s, a great development in molecular biology techniques has been observed. The popularization of PCR, traditional (Sanger) DNA sequencing and NGS techniques enabled unprecedented progress in many fields of biology, including phylogenetic studies. In the case of angiosperms, the general framework of its phylogeny as well as analyses of specific lineages within that group of plants were initially built based on genes derived from chloroplasts (*atpB, matK, rbcL*) and the nuclear (18S rDNA) genome [81,82,83]. However, despite the huge progress that was made in this field, there were still some lineages whose status or mutual phylogenetic relations were not clear. As a result, the researchers’ attention turned toward the third DNA-containing organelle, i.e., mitochondrion, and the potential of mitochondrial genes to solve the phylogenetic issues at different levels of plant systematics was confirmed [84,85,86]. In the present study, phylogenetic relationships among selected representatives of order Caryophyllales were reconstructed based on sequences of 18 mitochondrial protein-coding genes. The reconstructed phylogeny appeared to be consistent with the taxonomic position of the studied species, showing the separate character of the family Caryophyllaceae and close relationships between all studied *Colobanthus* species, with *C. lycopodioides* sharing the least similarity with other representatives of that genus. Comparison of the phylogenetic tree constructed for eight *Colobanthus* species based on sequences of 18 shared genes (the first variant of analysis) with the phylogenetic tree built on the sequences of 26 protein-coding genes (the second variant of the analysis) revealed their identical topology. This observation, together with the high support rates noted for both phylogenetic trees, proved the reliability of the presented results even for the variant of the analysis in which the reduced number of mitochondrial genes was applied. The presented here phylogenetic tree for *Colobanthus* species constructed based on mitochondrial genes and previously published trees which were generated based on 73 chloroplast genes [38] and two plastid intergenic spacers and nuclear sequence (*trnQ-rps16*, *atpB-rbcL,* and ITS, respectively; [87]) showed consistent topology. Taking into account the still unresolved status of many *Colobanthus* species, mitochondrial genes appeared as a valuable alternative for plastid and nuclear markers for determining the phylogenetic relationships within that group of plant species. Moreover, several potential problems of mitochondrial genes which might be observed during the phylogenetic study (i.e., horizontal transfer, RNA editing, or rate heterogeneity among lineages; for a review, see [86]) showed here no serious negative effects on the phylogenetic reconstruction of *Colobanthus* species. This observation confirmed that the mitochondrial genes are valuable, but still underutilized, DNA markers useful for resolving not only relationships between genera, families or higher taxa across angiosperms, but also at the interspecific level.

## 5. Conclusions

The current paper reported for the first time complete sequences of 26 protein-coding genes shared by eight *Colobanthus* species. Among these genes, one can distinguish 23 representatives of so-called core mitochondrial genes (for angiosperm), which are directly associated with the primary function of that organelle, and three other genes encoding sequences for components of the small unit of the ribosome, i.e., representing a facultative set of mitochondrial genes. The lack of *atp8* among genes for ATP synthase and the absence of sequences for transfer and ribosomal RNAs as well as most of the genes associated with the small and large subunit of ribosome or succinate dehydrogenase does not mean that they were not present in *Colobanthus* mitogenomes but rather that we were not able to obtain complete sequences for these genes for studied *Colobanthus* species from our sequencing data. Although the molecular data presented here could be considered as a preliminary study toward knowing the mitochondrial genomes of the genus *Colobanthus*, they have already proved to be a valuable source of information on the genetic diversity and evolution of this plant group. Moreover, they appeared to be concordant with previously reported information on the composition of mitochondrial genomes of *Silene*, which is another genus from the Caryophyllaceae family. The obtained results showed that there is a strong need to continue the studies on mitochondrial genomes of genus *Colobanthus*, studies which will provide us with complete sequences of the mitochondrial genomes, as only such data will enable a broader look at the evolution of mitochondrial genomes within the genus *Colobanthus* and Caryophyllaceae family.

## Figures and Tables

**Figure 1 genes-13-01060-f001:**
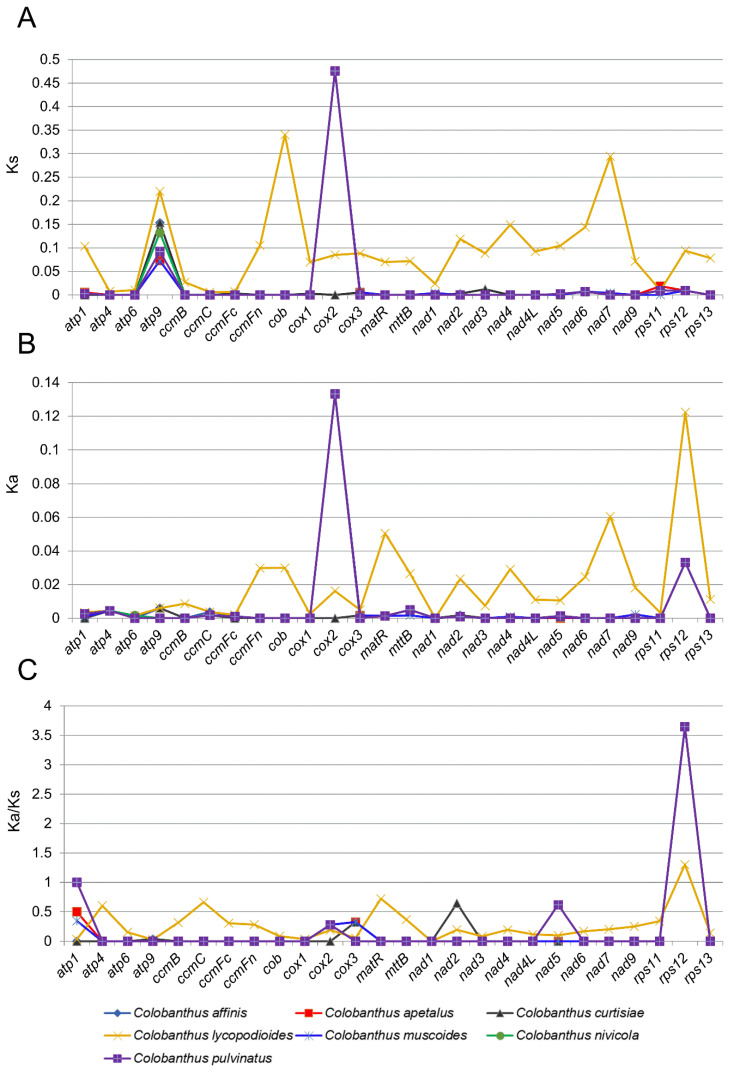
The evolution and dynamic of mitochondrial protein-coding sequences between eight species representing genus *Colobanthus*. The mt genome of *C. quitensis* was set as a reference. (**A**) synonymous (Ks) substitution rates; (**B**) non-synonymous (Ka) substitution rates; (**C**) gene-specific Ka/Ks ratios.

**Figure 2 genes-13-01060-f002:**
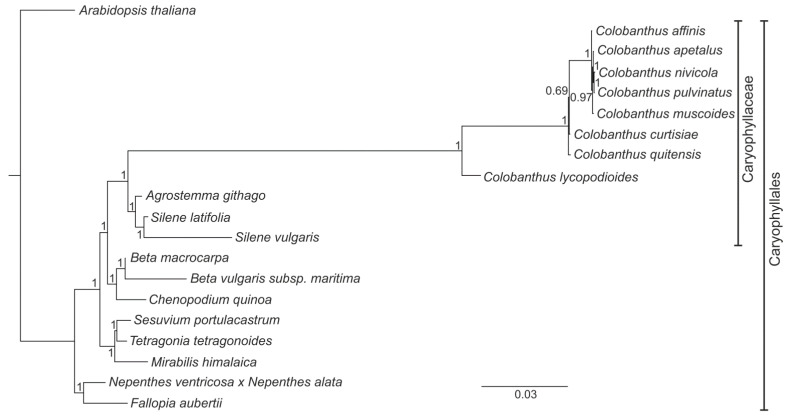
The phylogenetic tree based on sequences of shared 18 protein-coding genes from 19 representatives of order Caryophyllales and *A. thaliana* (outgroup) using Bayesian posterior probabilities (PP). Bayesian PP are given at each node.

**Figure 3 genes-13-01060-f003:**
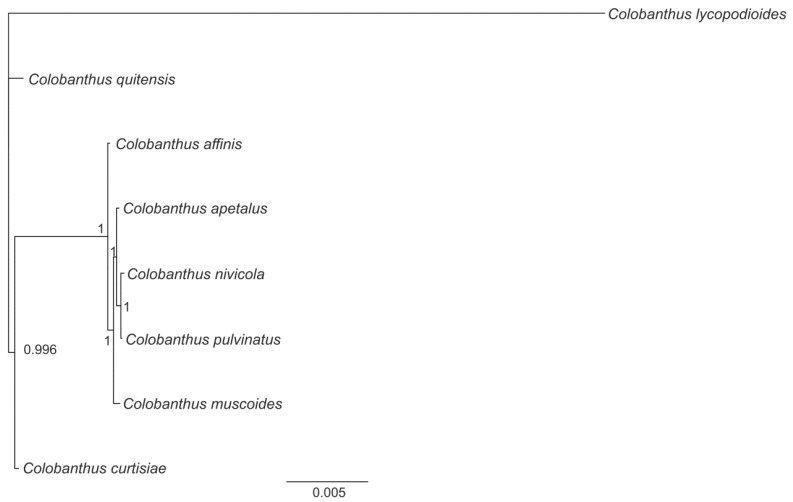
Phylogeny reconstruction between eight *Colobanthus* species based on sequences of shared 26 protein-coding genes using Bayesian posterior probabilities (PP). Bayesian PP are given at each node.

**Table 1 genes-13-01060-t001:** GenBank accession numbers for mitochondrial genomes used as a sources of protein-coding genes applied in phylogenetic analyses. Species list arranged alphabetically.

Accession Number	Species
NC_008285	*Arabidopsis thaliana*
NC_057604	*Agrostemma githago*
NC_015994	*Beta macrocarpa*
NC_015099	*Beta vulgaris subsp. maritima*
NC_041093	*Chenopodium quinoa*
MW664926	*Fallopia aubertii*
NC_048974	*Mirabilis himalaica*
NC_039531	*Nepenthes ventricosa × Nepenthes alata*
MN683736	*Sesuvium portulacastrum*
NC_014487	*Silene latifolia*
JF750427	*Silene vulgaris*
MW971440	*Tetragonia tetragonoides*

## Data Availability

The raw sequencing data obtained during the project realization were published in GenBank (NCBI) under project accession PRJEB52939. Additionally, the sequences of mitochondrial protein-coding genes were submitted as Appendix A.

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
