# Peer review of "Molecular Diversity and Phylogeny Reconstruction of Genus *Colobanthus* (Caryophyllaceae) Based on Mitochondrial Gene Sequences"

_genes, 2022, doi:10.3390/genes13061060_

Round 1
Reviewer 1 Report
Introduction: the largest plant mitochondrial genome is from Larix, and is 11.7 Mbp. The citation is Putintseva et al 2020. DOI: 10.1186/s12864-020-07061-4.
Discussion: In line 365, you talk about angiosperm mitogenome size however Larix is a gymnosperm. This sentence fits better in the introduction, where you talk about plants.
Introduction. Line 102: Beata macrocarpa must be replaced by Beta macrocarpa.
Results: in line 242 you say: " Three insertions and two deletions were observed in multiple alignments of the rps12 gene, which shortened the total length of the sequence in the mitochondrial genome of C. lycopodioides by nine nucleotides in comparison to other Colobanthus species." I consider that it is necessary to re-write this sentence to be more clear. What do you mean by: "multiple alignments? Do some reads support the deletion and other reads don't? It is possible that there are more than one copy in the mitochondrial genome and one copy has the deletion and then another does not?.
Did you find some indication of multiple copies of some genes? or by contrary, all reads mapped without mismatches to the consensus?
I consider it would be better to create contigs from reads and then, look for genes in these contigs.
M&M
Line 157: what do you mean by contigs? Did you do an assembly after mapping?
Due to the PREP suite, the web service will be ending by June I would recommend trying the Deep red-Mt prediction service which is available for download from GitHub here: [Deepred-Mt github]. The Deep red -Mt has proven to be better than PREP because instead of comparing your sequencing with a database, the Deep red- mt uses a neural network that takes into account the environment of the cytidine.
Figure 1. To better understanding of the chart, you should write the y axis legend net to the y-axis and not only in the paragraph under the Figure.
Regarding the gene search strategy, my advice is asemble the reads in contigs and search the genes in the contigs. So you will be able to find duplicated genes and genes that woulb be in te mtDNA in Colobanthus species and not in Silene.
Line 536: "The lack of atp8 among genes for ATP synthase and the absence of sequences for transfer and ribosomal RNAs as well as the most of genes associated with the small and large subunit of ribosome or succinate dehydrogenase does not mean that they were not present in Colobanthus mitogenomes, but that we were not able to obtain
complete sequences for these genes for studied Colobanthus species from our sequencing data." I believe that the absence of some genes does not depend of you sequencing data, but the strategy. The read depth of mitocondrial genes has proven be equal for all genes. I consider that would be very interesting know the gene content of the genus Colobanthus.
Author Response
Dear Reviewer,
thank you very much for your great effort and all valuable suggestions concerning our manuscript. We are sending our reply in the attached document.
Regards
Piotr Androsiuk

Reviewer 2 Report
The manuscript « Molecular diversity and phylogeny reconstruction of genus Colobanthus (Caryophyllaceae) based on mitochondrial gene sequences” by Androsiuk et al. is a solid, well-written study of this insufficiently characterised group of plants, which proposes to establish phylogeny of the Colobanthus genus by leveraging its mtDNA diversity. The authors obtained complete sequences of 26 mtDNA-encoded proteins and compared them among themselves and with those of more-or-less distantly related species. This enabled them to propose a phylogeny consistent with previous analyses involving nuclear and chloroplast sequences. They also made interesting predictions regarding the substitution rates and RNA editing in these genes. I only have a few minor points:
1. When one uses total DNA as starting material, there might be a danger that one mistakes mitochondrial pseudogenes located in the nuclear genome (i.e. NUMTs) for genuine mtDNA sequences. The authors should provide reasoning why they think the genes they have sequenced are truly located on mtDNA.
2. Why did the authors choose C. quitensis as reference for their Ka/Ks analysis? How would the outcome (and the authors’ conclusions) change if they used the most basal C. lycopodioides species as a reference? Can one imagine a “consensus reference” that would reflect a certain “average state” equally distant from all examined Colobanthus species?
3. In general, the manuscript would benefit from a bit of extra punctuation to facilitate reading. For instance, on l. 19, I would definitely insert a comma in “…from chloroplast and nuclear genes, also mitochondrial sequences revealed…”
4. Throughout the manuscript: Substitute “sharing” for “shearing”.
5. Throughout the manuscript: A correct term for rps proteins is “proteins of the small subunit of the ribosome” or “components of the ribosomal small subunit”.
6. L. 102: “Beta macrocarpa”.
7. L. 113: “26 mitochondrial protein-coding genes”.
8. L. 140: “growth rate”.
9. L. 217: The information about the maximum read length is trivial, given that sequencing was performed with 150-nt paired-end reads. It could be more informative to provide statistics about the median read length for the reads used to reconstruct the 26 mitochondrial genes: this would permit to appreciate their coverage depth.
10. L. 239: “rps12”?
11. L. 243: The authors mention multiple alignments without showing them. I would definitely include them as supplementary figures.
12. Table S2: There is a forgotten “8168” in the cell Q69 of this table.
13. Ll. 338-342: “clade”.
14. Ll. 364-366: This statement is somehow at odds with the one the authors make in the Introduction (ll. 68-70).
15. L. 402: “did not follow the above-mentioned rules”.
16. Ll. 509-510: “sharing the least similarity”.
Author Response
Dear Reviewer 2,
thank you very much for your great effort and all valuable suggestions concerning our manuscript. We are sending our reply in the attached document.
Regards
Piotr Androsiuk
